# Research on Prediction of TBM Performance of Deep-Buried Tunnel Based on Machine Learning

**Tianhui Ma** [1] , **Yang Jin** [2], **Zheng Liu** [1],*** **and Yadav Kedar Prasad** [1]

1   State Key Laboratory of Coastal and Offshore Engineering, Dalian University of Technology, Dalian 116024, China; tianhuima@dlut.edu.cn (T.M.); yadavkedar19@mail.dlut.edu.cn (Y.K.P.)
2   ZheJiang Poly Real Estate Development Co., Ltd., Hangzhou 311215, China; a1115139591@163.com
*   Correspondence: 32006019@mail.dlut.edu.cn

**Abstract:** Based on the relevant data in the construction process of the south of the Qinling tunnel of the Hanjiang-to-Weihe River Diversion Project, this article obtains the main influencing factors of the tunnel boring machine (TBM) performance of the deep-buried tunnel. According to the characteristics of deep-buried tunnel excavation, the random forest algorithm is used to select the features of the factors affecting the TBM penetration rate, and the four factors with large influence weights including total thrust, revolutions per minute, uniaxial compressive strength and volumetric joint count, are used as TBM penetration rate prediction models input parameters, which can improve the prediction accuracy and convergence speed of the model, and enhance the engineering practicality of the prediction model. Three types of TBM penetration rate prediction models are established: multiple regression model (MR), back propagation neural network model (BPNN) and support vector regression model (SVR). The prediction accuracy of the three models is compared and analyzed. The BPNN prediction model exhibits better prediction performance and generalization ability than the multiple regression model and SVR model, which manifest higher prediction accuracy and prediction stability.

**Keywords:** tunnel boring machine; penetration rate prediction; machine learning; deep-buried tunnel; feature selection

## 1. Introduction

Since the beginning of the 21st century, the world has been facing the challenges of water shortages, excessive population growth, energy crises and many other issues. The demand for water conservancy projects, underground transportation, deep-energy exploitation and so on is growing. Therefore, underground engineering has become an important development direction of infrastructure construction in the world, and tunnel construction is the key control point of these major projects [1].

Due to the wide application of the TBM in tunnel construction and the practical needs of construction projects, many experts and scholars at home and abroad have researched and developed many TBM performance prediction models since the 1970s. Forecasting research can be divided into two categories: theoretical research and empirical model research.

Most of the theoretical model research is based on laboratory tests and numerical simulations, or semi-theoretical research based on engineering knowledge, and then it carries out the prediction of the TBM performance. Sanio [2] thought that the rock breaking of the TBM hob is caused by the tensile failure of rock, and based on this assumption, the performance prediction formula of the TBM hob in the stratified and schistose rock mass is proposed. Hughes [3] explored the influence of hob size on the TBM penetration rate. Boyd [4] proposed the prediction formula of the TBM penetration rate based on rock mass specific energy, cutterhead power, tunnel cross-sectional area and the machine efficiency factor by using the dimensional analysis method. Ozdemir [5] and Rostami [6,7] proposed

and improved the famous Colorado school of mines (CSM) model. The model summarized a large number of laws through the full scale linear cutting machine to obtain the hob load and combined this with the specific rock mass parameters and mechanical parameters to obtain the prediction formula of the TBM penetration rate.

The theoretical analysis based on laboratory tests and numerical simulation is helpful to further reveal the rock breaking mechanism of the TBM hob. However, TBM construction is a systematic construction method integrating tunnel excavation, design and support, so the research on the TBM construction site is the mainstream research direction of TBM performance prediction.

In addition, a variety of empirical models are used to predict the TBM driving speed. Based on the geological data and the TBM performance parameters of eight TBM tunnels, Farmer et al. [8] proposed that the rock tensile strength ($\sigma_t$) and average thrust of cutterhead ($F_n$) should be used to calculate the penetration. Based on 112 groups of rock mass parameters and TBM performance parameters of five TBM tunneling projects in Italy, Innaurato et al. proposed a method to calculate the TBM penetration rate by using the rock structure score (RSR) and uniaxial compressive strength (UCS). Gong and Zhao [9] quantitatively analyzed the influence of the uniaxial compressive strength (UCS), rock brittleness index ($B_i$), number of unit volumetric joints ($J_v$) and angle between joint surface and tunnel axis ($\alpha$) on penetration based on the relevant mechanical-rock parameters collected during the TBM excavation of the Deep Tunnel Sewerage System (DTSS) Project in Singapore. Combined with the above parameters, the prediction formula of the specific rock mass drivability index was proposed by using a non-linear regression analysis. Du Lijie et al. [10] analyzed the relevant data of TBM tunneling in a granite lithology in Northeast China, and the research showed that the penetration index (FPI) had a strong correlation with uniaxial compressive strength (UCS), rock integrity coefficient ($K_v$), and cutterhead thrust ($F$) and established a multiple regression relationship. Luo Hua and Chen Zuyu et al. [11] divided TBM cutterhead rock breaking into three stages: extrusion, initiation and breaking. Combined with the relevant data of the TBM cutterhead crushing stage in the TBM section of the Jilin Yinsong Project, the prediction model of the TBM penetration rate was established with uniaxial compressive strength (UCS), intactness index of rock mass ($K_v$) and cutterhead thrust ($F$) as parameters.

With the more and more extensive application of the TBM, TBM related research has entered the era of big data. With the continuous development of machine learning theory and the emergence of 5G technology, the application of machine learning in TBM tunneling has a broad prospect.

Liu et al. [12] established the prediction model of rock mass parameters during TBM excavation by using the intelligent algorithm of single target stacking (SST) and improved support vector regression (SVR). The model was based on 180 sets of data sets of the Jilin Yinsong Project. The improved SVR algorithm showed higher prediction accuracy for rock mass parameters such as uniaxial compressive strength (UCS) and the rock mass brittleness index (BI). Goh et al. [13] proposed to use the multiple adaptive regression spline method (MARS) to predict the maximum ground settlement of the earth pressure balance shield tunnel. The model was established by using 148 sets of data from three independent earth pressure balance shield tunnel projects in Singapore. Compared with other methods such as the artificial neural network (ANN) and relevance vector machine (RVM), the model is simple to operate and easy to calculate the input rate of target contribution. Hassanpur et al. [14] developed a prediction model of the TBM jamming risk based on the Bayesian network (BN) and error back propagation neural network (BPNN) algorithm to predict the jamming risk index ($J_r$) to evaluate the design thrust value more reasonably in the design stage of TBM equipment. Zhang Na et al. [15] developed the intelligent control system of TBM tunneling parameters, which uses the combination of artificial neural network (ANN), support vector machine (SVM) and a least square regression to realize the prediction of TBM tunneling parameters. The system has been preliminarily applied in the TBM construction of the Jilin Yinsong Project and has certain feasibility.

In this paper, machine learning is used as the main technical means, relying on the south of the Qinling tunnel of the Hanjiang-to-Weihe River Diversion Project to carry out the relevant research. Based on a theoretical analysis and on-site actual data, the relevant methods of machine learning are used to develop the TBM tunneling efficiency prediction model of the deep-buried tunnel. By establishing the database of TBM performance in the south of the Qinling tunnel of the Hanjiang-to-Weihe River Diversion Project, the TBM mechanical parameters (thrust of cutterhead, torque of cutterhead, cutting speed, etc.) and rock mass parameters (uniaxial compressive strength, number of unit volumetric joints, angle between joint surface and tunnel axis, etc.) which affect TBM tunneling efficiency are quantitatively analyzed. Based on the quantitative analysis, the main influencing factors of TBM driving efficiency are obtained. After that, there are many influencing factors of the TBM penetration rate in the database of TBM performance. The random forest (RF) algorithm is used to select the characteristics of the influencing factors of the TBM penetration rate, and the factors that have a great influence on the TBM penetration rate are screened out. The input parameters are determined for the TBM penetration rate prediction model based on machine learning. Finally, based on the input parameters analyzed above, three prediction models of TBM penetration rate are established, which are multiple regression (MR), BP neural network (BPNN) and support vector regression (SVR). The prediction effect of the model is evaluated by the mean square error of the model and the absolute error and relative error of the prediction samples. It is concluded that the BPNN model has the best prediction effect.

## 2. Analysis of Influencing Factors and Feature Selection of TBM Performance

### 2.1. Establishing TBM Driving Efficiency Database

In order to analyze the TBM performance in the south of Qinling tunnel of the Hanjiang-to-Weihe River Diversion Project, the TBM performance database was established based on the data collected from the construction site. As shown in Table 1, the database is mainly composed of two categories: the first category is TBM mechanical parameters. By sorting out the TBM automatically recorded driving data and combining it with TBM construction time records, the corresponding operation parameters of TBM operation data and mileage number are obtained, mainly including cutterhead thrust (TF), penetration (P), torque (T), rotational speed (RPM), penetration rate (PR) and TBM utilization ratio (U), etc.

**Table 1.** Structure of the TBM performance database.

| TBM Mechanical Parameters | Rock Mass Parameters |
|---|---|
| Cutterhead thrust TF (kN) | Rock uniaxial compressive strength UCS (MPa) |
| Penetration $P$ (mm/r) | Rock abrasion resistance index $A_b$ ($10^{-1}$ mm) |
| Torque $T$ (N·m) | Specific work of chiseling $a$ (J/cm$^3$) |
| Rotational speed RPM (r/min) | Number of rock volume joints $J_v$ (pieces/m$^3$) |
| Rotational speed PR (m/h) | Angle between joint surface and tunnel axis $\alpha$ (°) |
| Construction speed AR (m/d) | Quartz content $q$ (%) |
| TBM utilization ratio $U$ (%) | P-wave velocity of rock mass $V_{pm}$ |
| Penetration index FPI (kN·r·mm$^{-1}$·m$^{-1}$) | P-wave velocity of rock block $V_{pr}$ |
| Number of tool change of cutterhead (N/d) | |

The second type is the rock mass parameters corresponding to each driving mileage of TBM, which are composed of a geological investigation in the early stage of construction; field investigation during construction and geophysical prospecting; and rock mass parameters that characterize the physical and mechanical properties of driving rock mass measured in the laboratory, mainly including rock uniaxial compressive strength (UCS), number of rock volume joints ($J_v$), angle between joint surface and tunnel axis ($\alpha$), rock abrasion resistance index ($A_b$), specific work of chiseling ($a$), and quartz content ($q$), etc.

Table 2 lists the basic information of various parameters in the TBM tunneling efficiency database collected in the 8521 m statistical section of the Lingnan Project in the TBM construction section.

**Table 2.** Basic information of parameters in TBM performance database.

| Parameters | Minimum Value | Maximum Value | Average Value | Standard Deviation |
|---|---|---|---|---|
| Cutterhead thrust TF (kN) | 3672 | 21,000 | 11,665.48 | 4436.8 |
| Penetration $P$ (mm/r) | 0.2 | 22 | 7.13 | 9.23 |
| Torque $T$ (kN·M) | 420 | 4645 | 2453 | 1046.84 |
| Rotational speed RPM (r/min) | 0.7 | 7.9 | 3.99 | 1.53 |
| Rotational speed PR (m/h) | 0.23 | 3.4 | 1.57 | 0.72 |
| Construction speed AR (m/d) | 0.7 | 25.3 | 8.3 | 4.29 |
| TBM utilization ratio $U$ (%) | 4.17 | 65.97 | 29.72 | 13.43 |
| Penetration index FPI (kN·r·mm$^{-1}$·m$^{-1}$) | 11.28 | 660.63 | 178.85 | 133.18 |
| Number of tool change of cutterhead (N/d) | 0 | 26 | 6.69 | 6.48 |
| Rock uniaxial compressive strength UCS (MPa) | 89.3 | 306.04 | 177.36 | 42.39 |
| Rock abrasion resistance index $A_b$ (10$^{-1}$·mm$^{-1}$) | 4.58 | 5.9 | 5.23 | 0.39 |
| Specific work of chiseling $a$ (J/cm$^3$) | 453.7 | 598.4 | 546.61 | 27.03 |
| Number of rock volume joints $J_v$ (pieces/m$^3$) | 2 | 27 | 12.88 | 4.76 |
| Angle between joint surface and tunnel axis $\alpha$ (°) | 0 | 90 | 57.27 | 18.03 |
| Quartz content $q$ (%) | 43.5 | 87.3 | 65.68 | 10.23 |
| P-wave velocity of rock mass $V_{pm}$ | 4200 | 6000 | - | - |
| P-wave velocity of rock block $V_{pr}$ | 3500 | 4898 | 3956 | 612.85 |

*2.2. Feature Selection of Influencing Factors of TBM Net Driving Speed Based on Random Forest*

2.2.1. Parameter Setting of Random Forest

There are two important parameters that need to be optimized in the random forest: the number of regression trees *n_estimators* in the sample data set extracted by the bootstrap method, and the maximum number of features max_features used on each node, which have four options, namely Auto/None, sqrt and 0.2. Liaw et al. [16] proposed that in order to balance the diversity of a single decision tree and the convergence speed of the algorithm, generally, max_features select sqrt, which can be calculated according to Formula (1).

$$\text{sqrt} = \sqrt{M} \tag{1}$$

where *M* is the number of input parameters of the model.

According to the previous analysis, there are 9 factors affecting the penetration rate of TBM. Therefore, sqrt is 3. In order to determine the best prediction model, the statistical section is divided into 170 driving units. According to the TBM performance database, the corresponding driving mileage, rock mass parameters and mechanical parameters of each driving unit are obtained. In total, 140 groups of data are used as training sets to train the RF model, and 30 groups of data are used as test sets to measure the prediction performance of the model. The prediction mean square errors of the random forest models with different combinations of the regression tree number *n_estimators* are compared. Firstly, the random forest models with a different number of regression trees *n_estimators* are numbered, and the value of *n_estimators* is from 1 to 1000 in 50 steps. The mean square error (MSE) is used to evaluate the prediction accuracy of the model, and the comparison results are shown in Figure 1. With the increase in the number of *n_estimators*, the mean square error

(MSE) of the RF model decreases gradually, and the overall error tends to be stable when *n_estimators* are greater than 350. Therefore, the number of *n_estimators* is determined; that is, the number of trees in the random forest is 350.

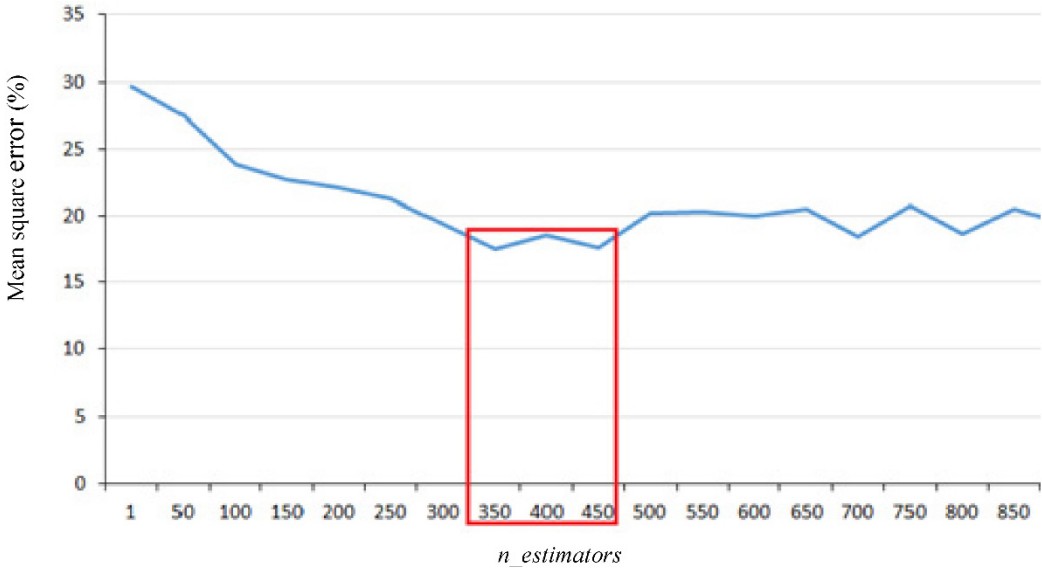

**Figure 1.** Variation trend of mean square error of RF model under various *n_estimators* quantity (*n_estimators* is the number of regression trees in the random forest. The red part represents the range of *n_estimators* when mean square error is minimum).

### 2.2.2. Feature Selection of Influencing Factors of TBM Penetration Rate Based on Random Forest

After the parameters of the random forest are determined, the weight of each parameter is generated on the whole sample set by using the random forest. The weight ranking table of each parameter is shown in Table 3. It can be seen from Table 3 that cutterhead thrust TF is the most important factor affecting the TBM penetration rate, and its weight proportion is 23.86%. The weight of the angle between joint surface and tunnel axis $\alpha$ is 7.97%. However, with the specific work of chiseling a rock abrasion resistance index $A_b$ and quartz content $q$, the influence weights of these three rock mass parameters on the TBM penetration rate are relatively small, and are 3.35%, 1.11% and 0.96%, respectively. According to the weight table analysis, the total weight of the four parameters TF, $J_v$, UCS and RPM is 75.99%, and they are the most important parameters affecting the penetration rate of the TBM.

**Table 3.** TBM penetration rate influencing factors' weight ranking.

| Input Parameter | Weight/% |
|---|---|
| Cutterhead thrust TF (kN) | 23.86 |
| Number of unit volumetric joints $J_v$ (pieces/m$^3$) | 19.71 |
| Uniaxial compressive strength UCS (MPa) | 17.89 |
| Rotational speed RPM (r/min) | 14.53 |
| Torque $T$ (kN·M) | 10.62 |
| Angle between joint surface and tunnel axis $\alpha$ (°) | 7.97 |
| Specific work of chiseling $a$ (J/cm$^3$) | 3.35 |
| Rock abrasion resistance index $A_b$ ($10^{-1}$·mm$^{-1}$) | 1.11 |
| Quartz content $q$ (%) | 0.96 |

Therefore, based on the random forest algorithm, the nine influencing factors of the TBM penetration rate are selected. Finally, four parameters are selected as the input parameters of the TBM penetration rate prediction model based on machine learning,

including cutterhead thrust TF, number of rock volume joints $J_v$, uniaxial compressive strength UCS and cutterhead speed RPM.

## 3. Prediction Model of TBM Penetration Rate Based on Machine Learning

### 3.1. Prediction Model of TBM Penetration Rate Based on Multiple Regression (MR)

3.1.1. Correlation Analysis of Penetration Index (FPI) and Rock Mass Parameters

Firstly, the correlation between the penetration index (FPI) and rock mass in the TBM driving efficiency database is explored, and the rock mass parameters which have a great influence on the FPI are obtained. The correlation between the penetration index (FPI) and rock mass parameters is shown in Figure 2. It can be seen from Figure 2 that there is a good correlation between the penetration index (FPI) and the uniaxial compressive strength (UCS). The correlation coefficient $R^2$ is 0.74; the correlation coefficient between the number of rock volume joints $J_v$ and the penetration index (FPI) also reaches 0.63; the correlation coefficients between the penetration index (FPI) and the angle between the joint surface and tunnel axis $\alpha$, the chiseling specific work a and the rock wear resistance index $A_b$ are 0.36, 0.38 and 0.13, respectively. According to the above analysis, the UCS of rock mass and the number of rock volume joints $J_v$ of rock mass, which represent the integrity of rock mass, will significantly affect the penetration rate of the TBM. Similarly, these two indexes show a good correlation with the penetration index (FPI), which is used to measure the boreability of the rock mass.

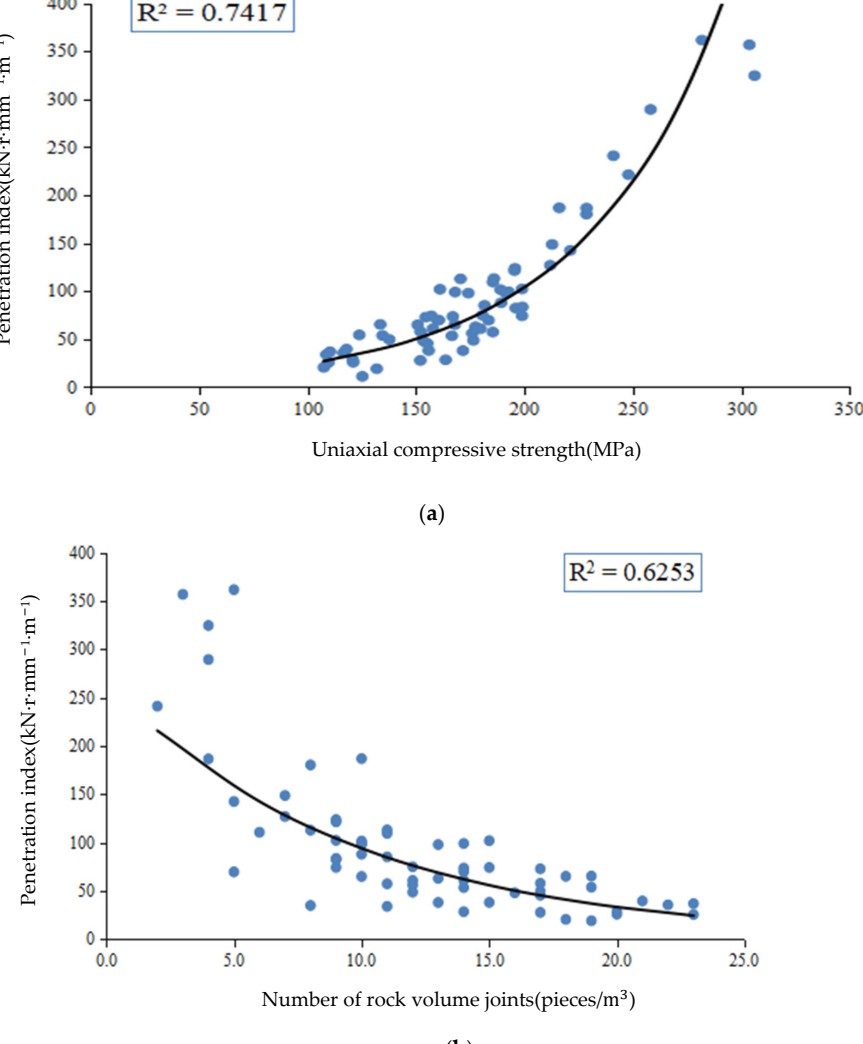

**Figure 2.** *Cont.*

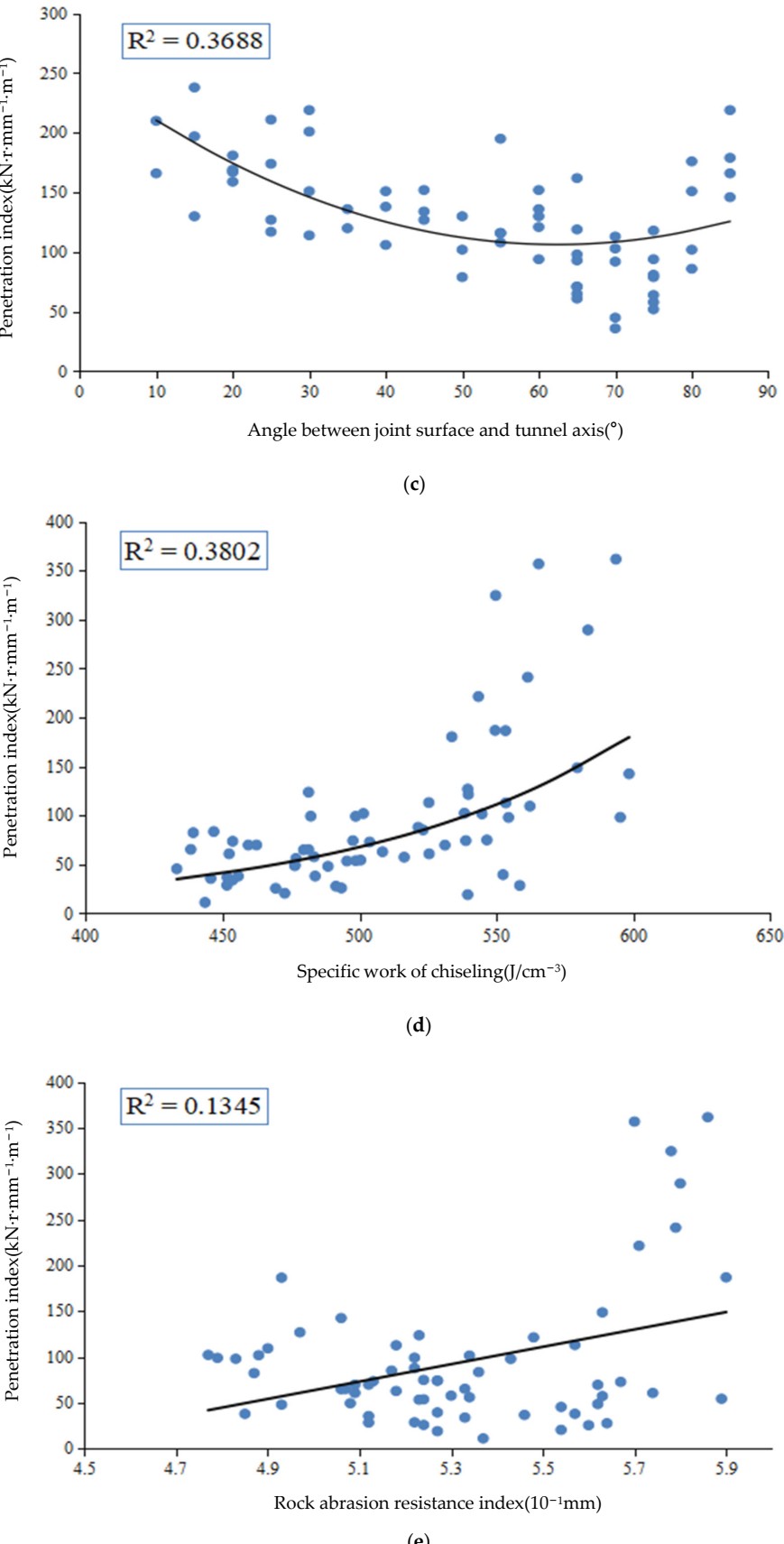

**Figure 2.** Correlation between field penetration index and rock mass parameters. (**a**) FPI-Uniaxial compressive strength UCS; (**b**) FPI-Number of rock volume joints $J_v$; (**c**) FPI-Angle between joint surface and tunnel axis $\alpha$; (**d**) FPI-Specific work of chiseling $a$; (**e**) FPI-Rock abrasion resistance index $A_b$.

3.1.2. Empirical Analysis of TBM Penetration Rate Prediction Model Based on Multiple Regression

Among the mechanical parameters of the TBM, the penetration index (FPI) includes two mechanical parameters which have the greatest impact on TBM performance, namely, cutterhead thrust TF and penetration P. There is a good correlation between them and the uniaxial compressive strength (UCS) and penetration index (FPI) in rock mass parameters, which can be used as a bridge connecting rock mass parameters and TBM mechanical parameters. Therefore, the penetration index (FPI) is used to explore the relationship between TBM mechanical parameters and rock mass parameters.

The uniaxial compressive strength (UCS) of rock mass and the number of rock volume joints $J_v$ representing the integrity of rock mass will be used as independent variables in the multiple regression analysis of the penetration index (FPI), and the empirical prediction formula based on the penetration index (FPI) will be established, namely Formula (2).

$$\text{FPI} = 1971.32 - 703.36 \ln J_v + 3.07\text{UCS} \tag{2}$$

where $J_v$ is the number of rock volume joints.

After the multiple regression analysis, the empirical formula shows good accuracy, and the goodness of fit $R^2$ is 0.76, which means that 76% of the change of the penetration index (FPI) can be explained by the uniaxial compressive strength (UCS) and the number of rock volume joints $J_v$.

After obtaining the penetration index (FPI), the expression of penetration P (3) can be obtained.

$$\text{P} = \frac{\text{TF} - f}{\text{FPI} \cdot \text{D}} \tag{3}$$

where TF is the cutterhead thrust, $f$ is the friction force to be deducted from the total thrust, and D is the cutterhead diameter. The weight of the TBM studied in this paper is 1300 t, which is calculated according to $f$ = 2600 kN.

Furthermore, according to the definition of penetration P (3), the expression of the penetration rate PR (4) can be obtained:

$$\text{PR} = \frac{\text{P} \cdot 60 \cdot \text{RPM}}{1000} = \frac{(\text{TF} - f) \cdot 60 \cdot \text{RPM}}{(1971.32 - 703.36 \ln J_v + 3.07\text{UCS}) \cdot 1000} \tag{4}$$

In this way, the prediction model of the TBM penetration rate based on multiple regression is obtained. The input parameters of the prediction model are cutterhead thrust (TF), rotational speed (RPM), uniaxial compressive strength (UCS) and number of rock volume joint $J_v$. Through the correlation of the penetration index (FPI) with rock mass parameters and mechanical parameters, it is verified that the random forest model is scientific and reasonable in ranking the weight of influencing factors of the penetration rate.

To evaluate the accuracy of the prediction model of a penetration rate based on multiple regression, the predicted value of the model is compared with the measured penetration rate value in the data set. The comparison results are shown in Figure 3. It can be seen from Figure 3 that the correlation coefficient between the predicted value and the measured value of the TBM penetration rate (PR) is 0.7, and the deviation between the predicted value and the measured value is small in the range of 0.5–1.5 m/h, which may be due to a large number of samples in this range included in the sample set. After the penetration rate is greater than 2 m/h, the deviation between the predicted value and the measured value is large. In general, the predicted value can reflect the variation law of the actual net driving speed PR of the TBM within a certain range, indicating that the empirical prediction formula of the penetration rate (PR) of the TBM determined by the method of multiple regression analysis can be applied to the preliminary prediction of the penetration rate of the TBM in the south of the Qinling tunnel of the Hanjiang-to-Weihe River Diversion Project.

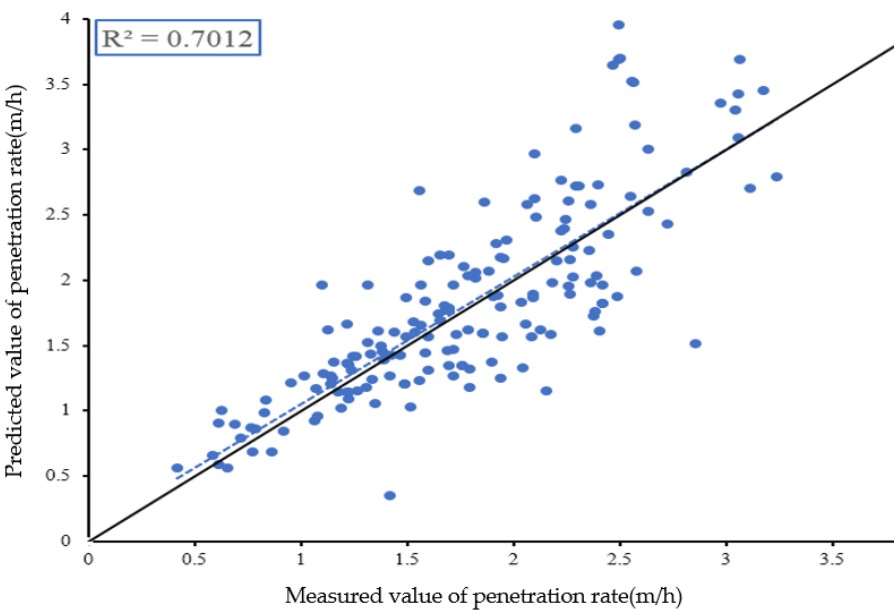

**Figure 3.** Comparison of predicted and measured value of penetration rate.

*3.2. Prediction Model of TBM Penetration Rate Based on BP Neural Network*

3.2.1. Principle of Back Propagation Neural Network Algorithm

The back propagation neural network (BPNN) was proposed by Rumelhart et al. [17] in 1986. It is a neural network that continuously modifies the connection weights between various neurons through the training method of error back propagation. The neuron model is shown in Figure 4. $f(x)$ is called the activation function, which enables the neural network to deal with nonlinear problems. $s_i$ is the input from the previous neuron, $w_{ij}$ is the connection weight between the two neurons and $\theta_j$ is the threshold of the neuron. If $Y_j$ is the output value of $s_i$ after passing through a neuron, the output signal $Y_j$ of the jth neuron can be calculated by Equation (5).

$$Y_j = f\left(\sum w_{ij}s_i - \theta_i\right) \tag{5}$$

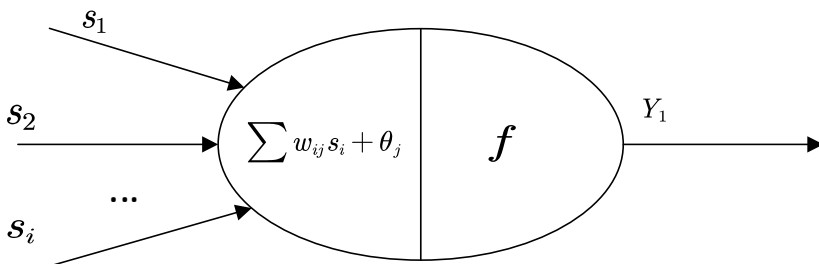

**Figure 4.** Neuron model.

The structure of the back propagation neural network is obtained by extending a single neuron, as shown in Figure 5. It includes three layers, namely the input layer, hidden layer and output layer.

For a training sample $(x, y) \in R^d \times R^l$, the corresponding output vector is $\hat{y} = (\hat{y}_1, \hat{y}_2 \ldots, \hat{y}_n)$. After each neuron connection calculation, the loss function on the output node can be expressed by the three most common loss functions: mean square error, mean absolute percentage error and mean absolute error. The specific expressions are in Equations (6)–(8).

$$\text{MSE} = \frac{1}{n}\sum_{j=1}^{n}\left(y_j - \hat{y}_j\right)^2 \tag{6}$$

$$\text{MAPE} = \frac{100\%}{n}\sum_{j=1}^{n}\left|\frac{y_j - \hat{y}_j}{y_j}\right| \tag{7}$$

$$\text{MAE} = \frac{1}{n}\sum_{j=1}^{n}\left|y_j - \hat{y}_j\right| \tag{8}$$

The essence of the back propagation neural network algorithm is gradient descent. When training the neural network, the iterative update formula of any parameter is:

$$w_{new} = w_{old} + \Delta w \tag{9}$$

The iterative process of the weight $w_{ij}$ from the hidden layer to the output layer is shown in Equations (10)–(16):

$$\Delta w_{ij} = -\eta \frac{\partial E}{\partial w_{ij}} \tag{10}$$

According to the chain derivation rule:

$$\Delta w_{ij} = -\eta \frac{\partial \beta_j}{\partial w_{ij}}\frac{\partial \hat{y}_j}{\partial \beta_j}\frac{\partial E}{\partial \hat{y}_j} \tag{11}$$

$$\frac{\partial \beta_j}{\partial w_{ij}} = b_i \tag{12}$$

$$\frac{\partial \hat{y}_j}{\partial \beta_j} = \hat{y}_j(1 - \hat{y}_j) \tag{13}$$

$$\frac{\partial E}{\partial \hat{y}_j} = (\hat{y}_j - y_j) \tag{14}$$

Therefore, after error back propagation, the change of the neuron connection weight $w_{ij}$ is:

$$\Delta w_{ij} = \eta \hat{y}_j(1 - \hat{y}_j)(\hat{y}_j - y_j) \tag{15}$$

Similarly, the change of the neuron connection threshold is expressed by Formula (16):

$$\Delta \theta_j = -\eta \hat{y}_j(1 - \hat{y}_j)(\hat{y}_j - y_j) \tag{16}$$

At this time, the error is propagated back to the hidden layer. Similarly, the connection weight $w_{ij}$ and the connection threshold from the hidden layer to the input layer can be calculated $\theta_{jo}$ It can be seen from the algorithm principle of the BPNN model that BPNN can learn and store a large number of input–output mode mapping relationships without knowing the mathematical expression of this mapping relationship in advance. Its learning rule is to use the steepest descent method to continuously correct the weight $w_{ij}$ and threshold between each neuron through back propagation error $\theta_j$. That is, the mapped expression makes the loss function of the BPNN model reach the minimum value, and finally outputs the prediction result.

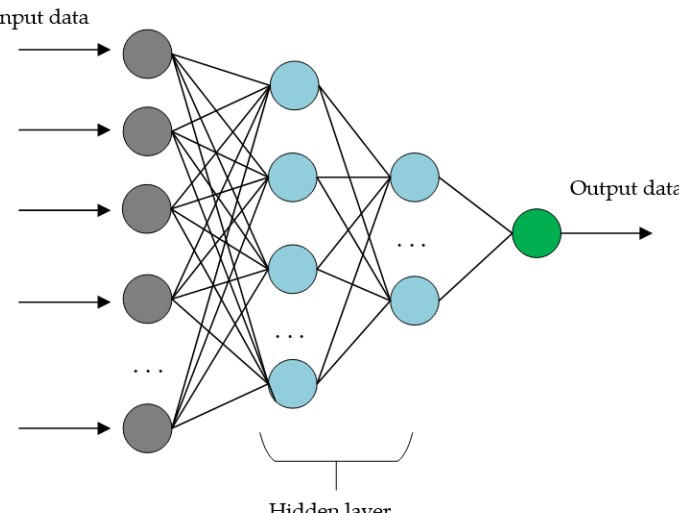

Input data

Output data

Hidden layer

**Figure 5.** Back propagation neural network structure diagram.

3.2.2. BPNN Model Structure and Parameter Setting

The structure of the BPNN model will significantly affect the accuracy and convergence rate of the prediction model. The input data are composed of four parameters screened by a random forest algorithm, namely thrust (TF), rotational speed (RPM), uniaxial compressive strength (UCS) and number of rock volume joint $J_v$. The output parameter is the penetration rate (PR) of the TBM. After the input and output layers are determined, the number of hidden layers and the number of hidden neurons need to constantly adjust the parameter settings to determine the best settings; otherwise, the prediction model will be overfitting or underfitting [18]. George [19] proposed that the BPNN of a single hidden layer can approximate a continuous nonlinear function within a limited number of hidden neurons, so the hidden layer is determined as 1. The number of hidden neurons is determined according to the empirical Formula (17) proposed by Matias [20].

$$K = \sqrt{a + b} + c \tag{17}$$

where $a$ and $b$ are the number of input and output nodes, $c$ is a constant between 1 and 10 and $K$ is the number of hidden neurons.

The numbers of nodes in the input layer and output layer are 4 and 1, respectively. Then, $K$ is a constant between 3 and 13. The selection of the activation function also has a crucial impact on the accuracy of the BPNN [21]. The common activation functions include tanh, sigmoid and ReLU functions. In order to determine the best prediction model, the statistical section is divided into 170 driving units, in which 110 groups of data are used as training sets to train the BPNN model, and 30 groups of data are used as test sets to measure the prediction performance of the model under different parameter settings. The number of hidden neurons and different combinations of the activation function of the BPNN are compared. BPNN models with different combinations of hidden neuron numbers and activation functions are numbered. A, B and C, respectively, mean tanh, sigmoid and ReLU functions selected as activation functions. Numbers represent the number of hidden neurons. The mean square error (MSE) and mean absolute percentage error (MAPE) are selected to evaluate the prediction accuracy of the model. The comparison results are shown in Figure 6. It can be seen from Figure 6 that when the activation function is the tanh function and the number of hidden neurons is 11, the BPNN model obtains the minimum mean square error and average absolute percentage error, and the prediction effect is the best. Accordingly, by adjusting the parameters, batch size, that is, the number of samples for a training, is set to 5; learning rate, that is, the learning rate, is set to 0.001.

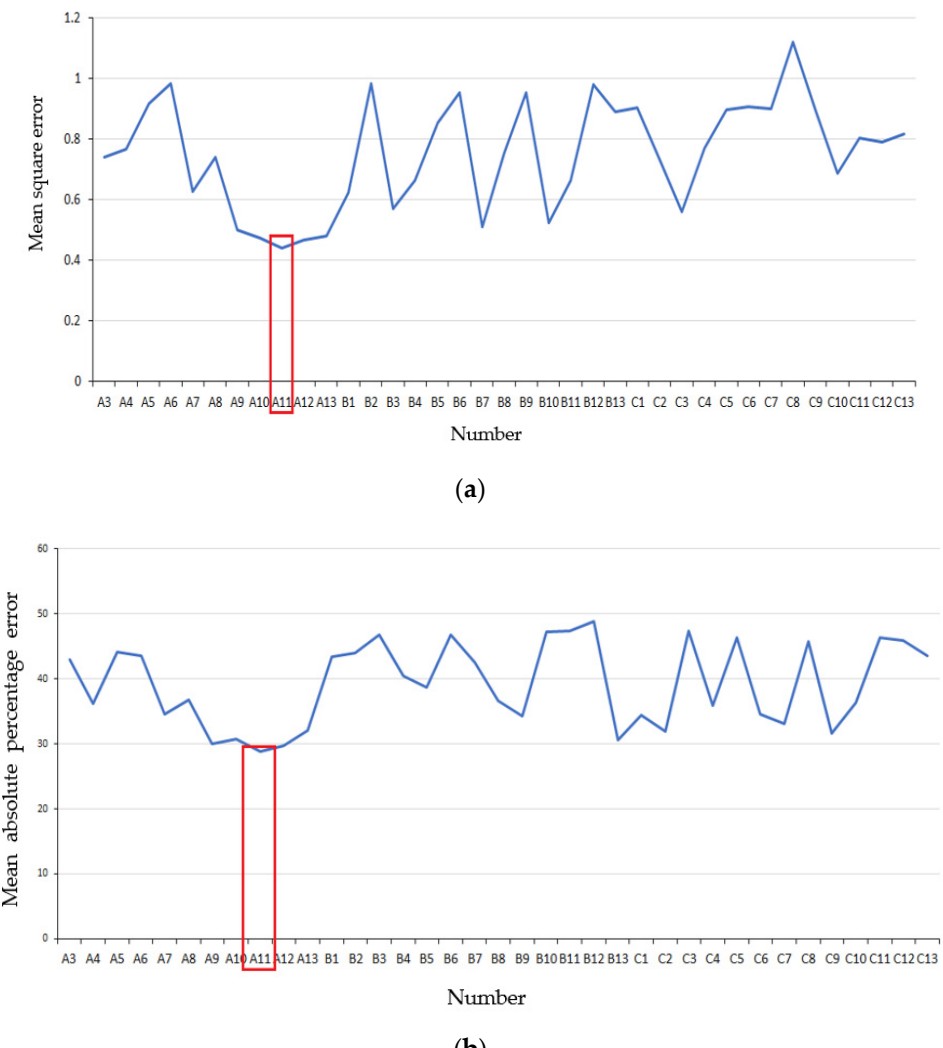

**Figure 6.** The trend of BPNN error variation under different parameter combinations (A, B and C, respectively, mean tanh, sigmoid and ReLU functions are selected as activation functions. Numbers represent the number of hidden neurons. The red part represents the parameter combination selected when mean square error is minimum in (**a**). The red part represents the parameter combination selected when mean absolute percentage error is minimum in (**b**)). (**a**) Mean square error, MSE; (**b**) Mean absolute percentage error, MAPE.

### 3.2.3. Empirical Analysis of TBM Penetration Rate Prediction Model Based on BPNN Model

After 110 data sets of 170 data sets are used as training sets to train the BPNN model, 30 data sets are used as test sets to measure the prediction performance of different parameter combinations, and the remaining 30 data sets are used as verification sets to measure the prediction performance of the model.

Validation by validation set, the mean absolute percentage error (MAPE) and mean square error (MSE) of the BPNN model are 22.95% and 0.3445, respectively. The comparison curve between the predicted value and actual value of 30 groups of samples is shown in Figure 7. The variation trend of absolute error Δ of each sample is shown in Figure 8, and the relative error δ of each sample is shown in Figure 9.

It can be seen from Figure 7 that the predicted value of the penetration rate based on the BPNN model can reflect the changing trend of the actual value. Except for sample number 6, the prediction performance of the predicted value between sample number 1 and 20 is better than that between sample number 20 and 30. It can be seen from Figure 8 that the absolute

error of 19 samples is less than 0.5 m/h, and the absolute error of sample 2 and 3 are 0.017 m/h and 0.021 m/h, respectively. The absolute error of 11 samples is more than 0.5 m/h, among which the absolute error of sample 22 and 23 is 1.27 m/h and 1.19 m/h, respectively. It can be seen from Figure 9 that the relative error of 16 samples is less than 20%, among which the relative errors of No. 2 and No. 7 samples are all 1.42%, and the relative error of 14 samples is more than 20%, among which the relative error of No. 6 sample is the largest, reaching 72%.

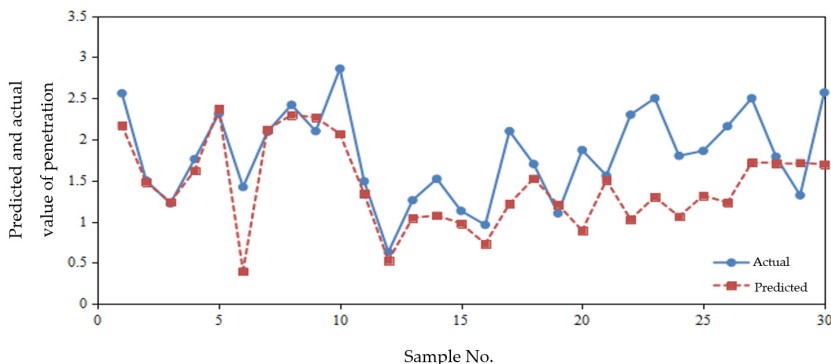

**Figure 7.** Comparison curve of the predicted value of penetration rate based on BPNN model and actual value of penetration rate (the horizontal axis represents the number of thirty groups of samples in the test set).

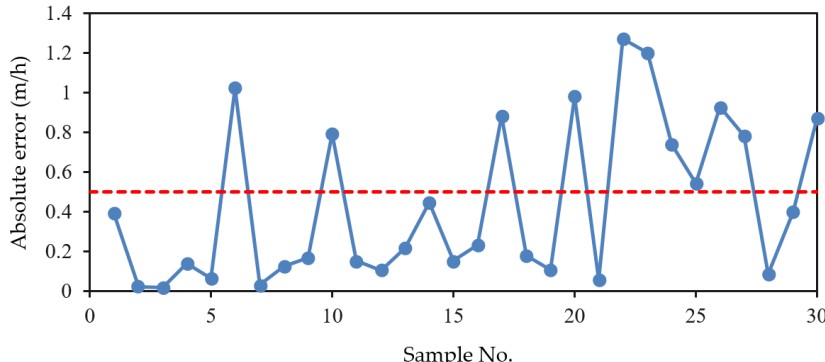

**Figure 8.** The absolute error variation curve of the predicted value of penetration rate based on BPNN model and actual value of penetration rate (the horizontal axis represents the number of thirty groups of samples in the test set).

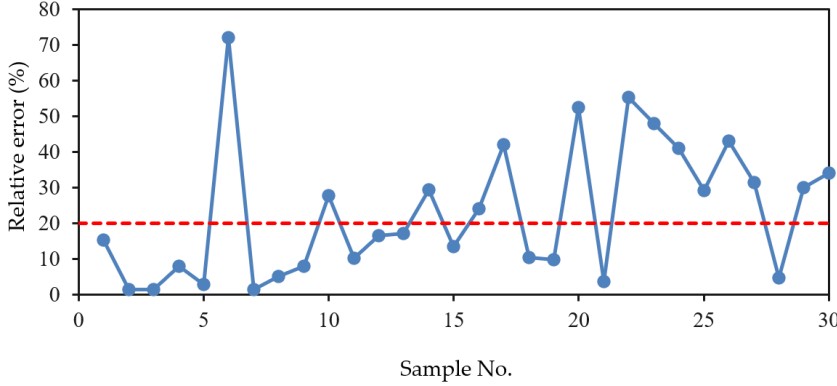

**Figure 9.** The relative error variation curve of penetration rate based on BPNN model and actual value of penetration rate. (The horizontal axis represents the number of thirty groups of samples in the test set).

### 3.3. Prediction Model of TBM Penetration Rate Based on Support Vector Regression

3.3.1. Principle of Support Vector Regression Algorithm

As a general method to solve the problem of high-dimensional function estimation, support vector regression is established based on the Vapnik chervonenkis (VC) theory [22]. If the dimension of VC is low, the expected error probability is also low, which indicates that the generalization ability is strong. Support vector regression is an optimization problem. First, define $\varepsilon$-insensitive loss function, minimize it, and then find the narrowest one containing most of the training data sets $\varepsilon$-insensitive zone. Therefore, according to the geometric characteristics of the loss function and the $\varepsilon$-insensitive zone, a multi-objective function is constructed.

Given a training set $\{(x_1, y_1), \ldots, (x_n, y_n)\} \subset R^M \times R$, where $R^M$ is the input space of the input feature $x_i$ and $y_i$ is the observed object, the goal is to find a function $f(x)$ that is within the deviation range of $\varepsilon$ from the actual target value $y_i$, and make it as gentle as possible. $f(x)$ can be expressed in linear form, as shown in Equation (18).

$$y = f(x_i) = \langle \mathbf{w_i}, \mathbf{x_i} \rangle + b \tag{18}$$

where $w_i$ represents the weight vector of the linear function whose unit length is at right angles to the hyperplane, and $b$ represents the threshold. A support vector regression approximates the data set with a linear function. The form of the linear function in the high-dimensional feature space is shown in Equation (19).

$$y = f(x) = \sum_{i=1}^{N} w_i(x_i) + b \tag{19}$$

where $(x_i)$ transfers the input vector to the feature space, that is, non-linear mapping from the input space $R^M$ to the high-dimensional feature space.

Using the $\varepsilon$-insensitive loss function in the error risk minimization with regularization, the support vector regression function can be established more conveniently. Only the samples in the $\varepsilon$-insensitive band have non-zero relaxation variables. Generally, if the predicted value is within the region, the loss is zero; if the prediction point is outside the insensitive band, the error is the difference between the prediction value and the insensitive band radius $\varepsilon$.

Loss function represents less than $\varepsilon$. There is no loss for the deviation, and the larger deviation will be subject to linear penalty [23], as shown in Formula (20):

$$L(\zeta) = \begin{cases} 0 & if |\zeta| < \varepsilon \\ |\zeta| - \varepsilon & \text{otherwise} \end{cases} \tag{20}$$

where the parameter $\varepsilon$ is equivalent to the approximate accuracy of the training data points. When the data points are within the range of $\pm\varepsilon$, it will not cause errors.

The support vector regression describes the function approximation problem as an optimization problem by finding the narrowest insensitive band centered on the surface and minimizing the prediction error, that is, the distance between the prediction and the expected output. The objective function is shown in Equation (21).

$$\min \frac{1}{2}\|w\|^2 \tag{21}$$

The goal of the support vector regression algorithm is to find the optimal linear hyperplane, and the optimal hyperplane can be guaranteed by comprehensively considering the regression error and flatness [23]. Combined with Formulas (22)–(25), this can be achieved by minimizing the objective function:

$$\min_{w,b} \left[ \frac{1}{2}\|w\|^2 + C\sum_{i=1}^{m} (\zeta_i^+ + \zeta_i^-) \right] \tag{22}$$

Constraints are (23)–(25):

$$f(x_i) - y_i \leq \varepsilon + \zeta_i^+ \tag{23}$$

$$y_i - f(x_i) \leq \varepsilon + \zeta_i^- \tag{24}$$

$$\zeta_i^+, \zeta_i^- \geq 0, \ i = 1, 2, \ldots, m \tag{25}$$

The constant $c > 0$ is a penalty parameter used to adjust the regression error and flatness weight. The relaxation variables $\zeta_i^+$ and $\zeta_i^-$ are the training errors calculated from the $\varepsilon$-insensitive loss function. Here, $\frac{1}{2}\|w\|^2$ represents the structural risk, which is used to control the smoothness or complexity of the function (regularization term); and $C \sum_{i=1}^{m} (\zeta_i^+ + \zeta_i^-)$ represents the empirical risk. Therefore, the support vector regression can be understood as the minimization of the structural risk and empirical risk.

Because the hyperplane of support vector regression is linear, the prediction accuracy of the linear equation may not be high for most nonlinear problems. This problem can be solved by mapping samples to high-dimensional space through the kernel function, and then the objective function can be transformed as shown in Formula (26) and constraint (27).

$$\max_{\alpha, \alpha^*} \sum_{i=1}^{m} [y_i(\alpha_i^* - \alpha_i) - \varepsilon(\alpha_i^* + \alpha_i)] - \frac{1}{2} \sum_{i=1}^{m} \sum_{j=1}^{m} (\alpha_i^* - \alpha_i)(\alpha_j^* - \alpha_j) K(x_i, x_j) \tag{26}$$

$$\sum_{i=1}^{m} (\alpha_i^* - \alpha_i) = 0 \quad \alpha_i^*, \alpha_i \in [0, C] \tag{27}$$

Among them, $\alpha_i$ and $\alpha_i^*$ are Lagrange multipliers, and $K(x_i, x_j)$ is the kernel function of input vector $x$. Its main function is to convert data points from low-dimensional space to high-dimensional space, so that data points can be divided by linear functions.

### 3.3.2. SVR Parameter Setting

In the support vector regression (SVR) algorithm, the kernel function determines the result of sample transformation. Therefore, the choice of kernel function is one of the important factors affecting the accuracy of the SVR model. The prediction accuracy and generalization ability of the support vector regression model also depend on the selection of loss function parameter $\varepsilon$ and penalty parameter $C$ [24]. Therefore, it is necessary to adjust and test $\varepsilon$, $C$ and $K(x_i, x_j)$ to obtain good prediction performance and the generalization ability of the SVR model.

The most commonly used kernel functions of the SVR are the radial basis function (RBF) and polynomial kernel function (Poly). The penalty parameter $C$ is generally between 0.1 and 100, and the loss function parameter $\varepsilon$ is generally between 0.01 and 0.1. To determine the best prediction model, the statistical section is divided into 170 driving units, in which 110 groups of data are used as training sets to train the SVR model, and 30 groups of data are used as test sets to measure the prediction performance of the model under different parameter settings. The SVR models with different combinations of kernel functions, penalty parameter $C$ and loss function parameter $\varepsilon$ are compared. The SVR models with different kernel functions, $C$ and $\varepsilon$ combination are numbered. A and B, respectively, mean RBF and Poly functions are selected as kernel functions. The value of $C$ is in the range of 0.1–100 in steps of 10; the value of $\varepsilon$ takes 0.05 as the step, takes the number between 0.01 and 0.1, and the number is 1–3. The mean square error (MSE) and mean absolute percentage error (MAPE) are used to evaluate the prediction accuracy of the model. The comparison results are shown in Figure 10. It can be seen from Figure 10 that when the kernel function is the RBF function, C is 10 and $\varepsilon$ is 0.1, the SVR model obtains the minimum mean square error and mean absolute percentage error, and the prediction effect is the best.

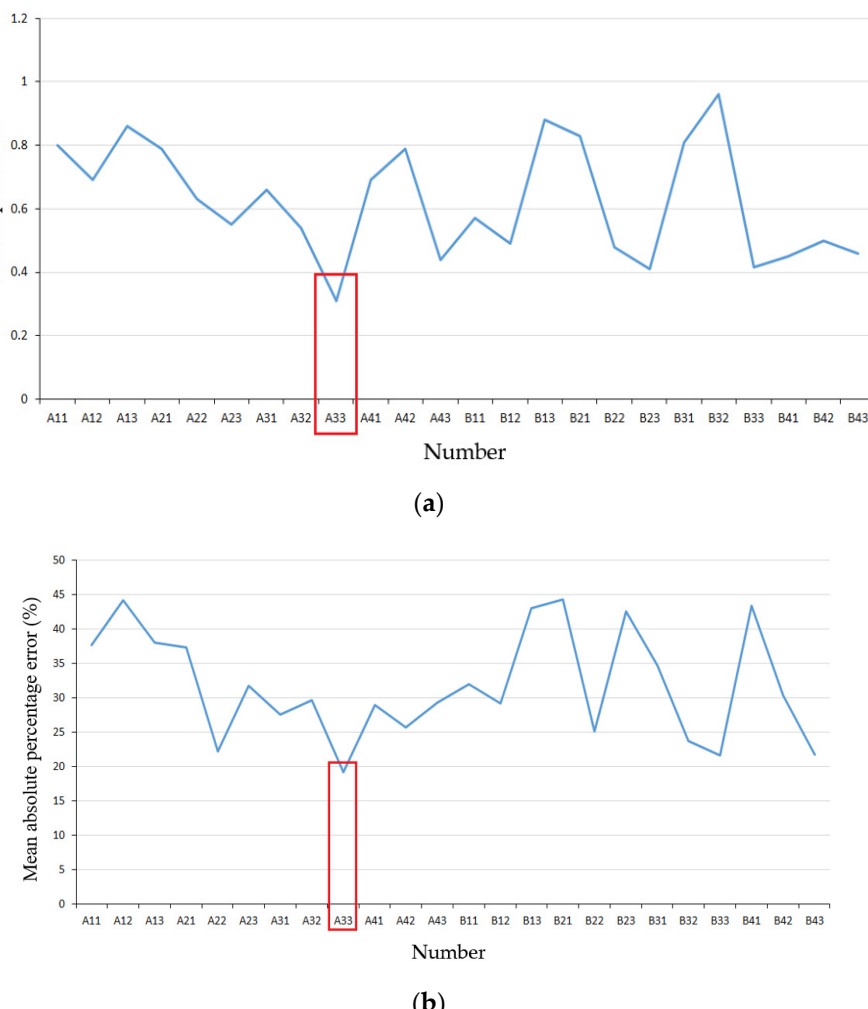

**Figure 10.** The trend of BPNN error variation under different parameter combinations (A and B, respectively, mean RBF and Poly functions are selected as kernel functions. Numbers represent the combination of different values of *C* and *ε*. The red part represents the parameter combination selected when mean square error is minimum in (**a**). The red part represents the parameter combination selected when mean absolute percentage error is minimum in (**b**)). (**a**) Mean square error (MSE); (**b**) Mean absolute percentage error (MAPE).

### 3.3.3. Empirical Analysis of TBM Penetration Rate Prediction Model Based on SVR Model

Similarly, 110 data sets of 170 data sets are used as training sets to train the SVR model, 30 data sets are used as test sets to measure the model prediction performance under different parameter settings and the remaining 30 data sets are used as verification sets to measure the model's prediction performance.

Validation by validation set, the mean absolute percentage error (MAPE) and mean square error (MSE) of the SVR model are 26.59% and 0.4085, respectively. The comparison curve between the predicted value and actual value of 30 groups of samples is shown in Figure 11. The variation trend of the absolute error Δ of each sample is shown in Figure 12, and the relative error δ of each sample is shown in Figure 13.

It can be seen from Figure 11 that the predicted value of the penetration rate based on the SVR model can reflect the changing trend of the actual value, and the prediction performance of the predicted value between sample numbers 1–5 is better. It can be seen from Figure 12 that the absolute error of 19 samples is less than 0.5 m/h, and the absolute error of No. 12 and No. 28 samples are 0.034 m/h and 0.05 m/h, respectively. The absolute error of 11 samples is greater than 0.5 m/h, among which the absolute error of the No. 22 and No. 23 sample is 1.23 m/h and 1.32 m/h, respectively. It can be seen from Figure 13

that the relative error of 15 samples is less than 20%, of which the relative error of No. 1 and No. 28 samples are 2.87% and 2.82%, respectively. The relative error of 15 samples is more than 20%, of which the relative error of the No. 6 sample is 65%.

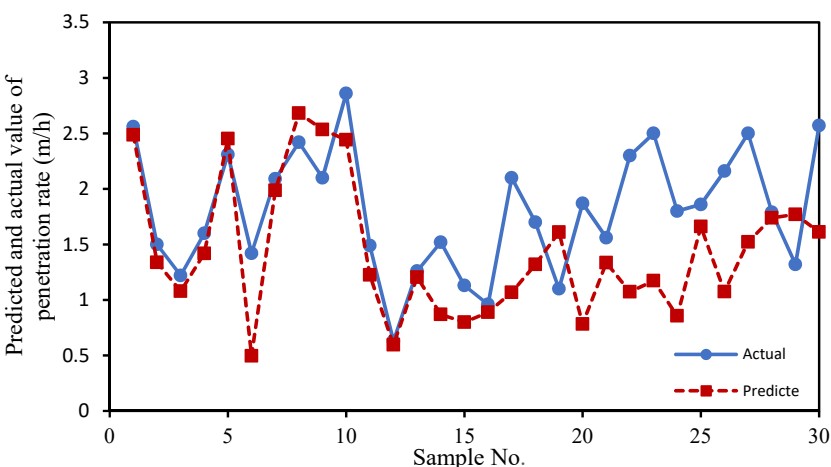

**Figure 11.** Comparison curve of predicted value of penetration rate based on SVR model and actual value of penetration rate. (The horizontal axis represents the number of thirty groups of samples in the test set).

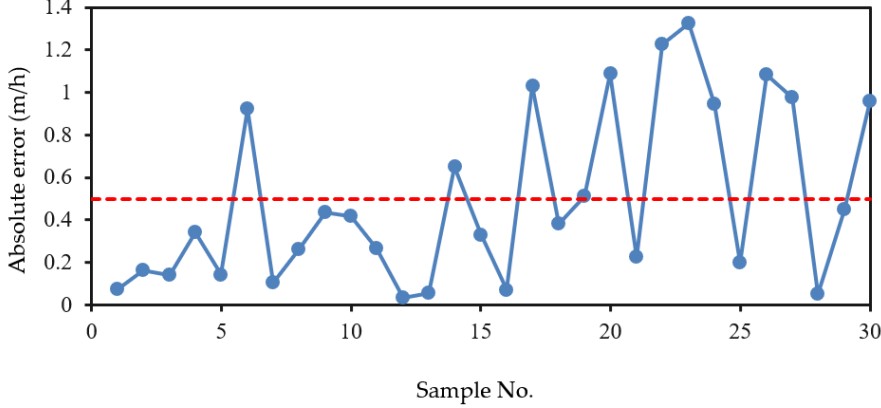

**Figure 12.** The absolute error variation curve of the predicted value of penetration rate based on SVR model and actual value of penetration rate (the horizontal axis represents the number of thirty groups of samples in the test set).

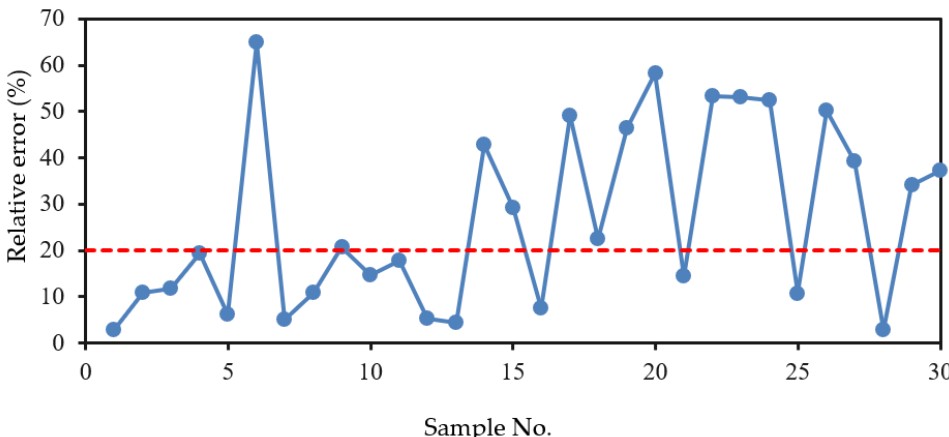

**Figure 13.** The relative error variation curve of a penetration rate based on SVR model and actual value of penetration rate (the horizontal axis represents the number of thirty groups of samples in the test set).

### 3.4. Comparative Analysis of Prediction Performance of Penetration Rate Prediction Models

In order to compare and analyze the prediction performance of the TBM penetration rate prediction models based on multiple regression, BPNN and SVR, the comparison curves between the predicted value of the TBM penetration rate and the field measured value of the three prediction models are drawn as shown in Figure 14, the absolute error change curve is shown in Figure 15 and the relative error change curve is shown in Figure 16.

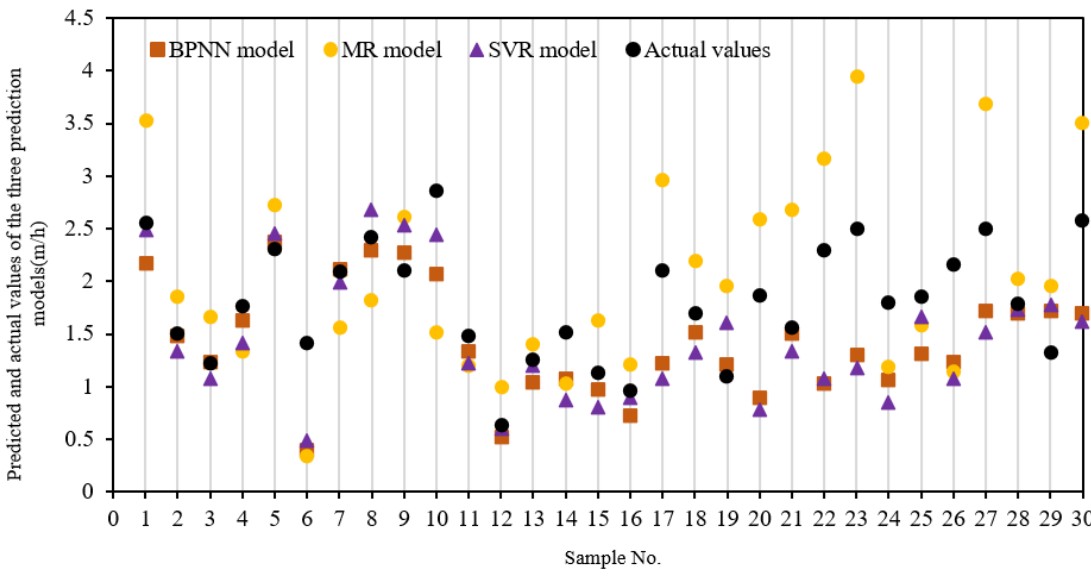

**Figure 14.** Scatterplot of the predicted and actual values of the three prediction models (the horizontal axis represents the number of thirty groups of samples in the test set).

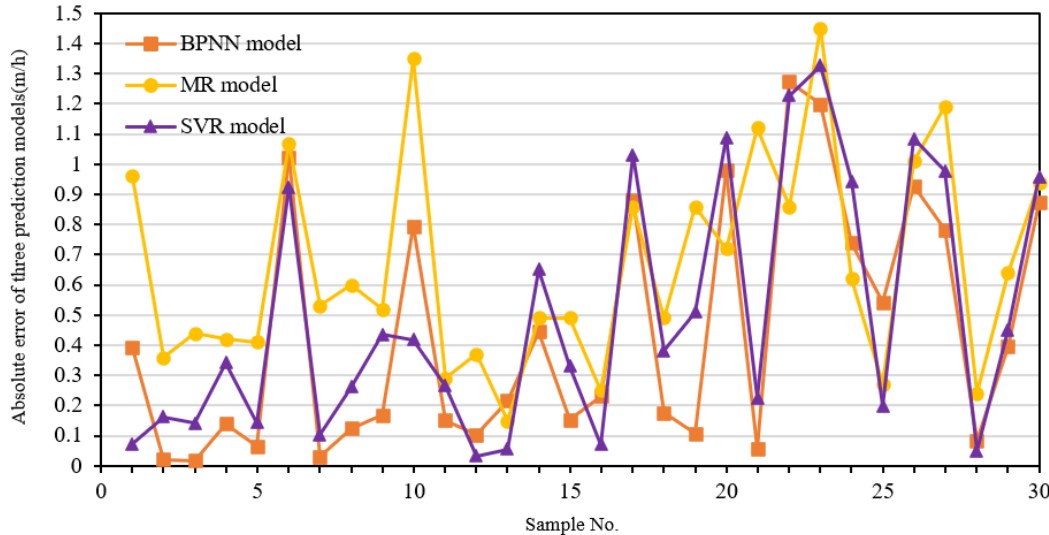

**Figure 15.** Comparison curve of absolute error between predicted value and actual value of three prediction models. (The horizontal axis represents the number of thirty groups of samples in the test set).

It can be seen from Figure 14 that both the BPNN model and SVR model can reflect the changing trend of the actual value of the TBM penetration rate, but the deviation between the predicted value and the actual value of the multiple regression prediction model is large, and the prediction ability to reflect the change trend of the actual value is poor.

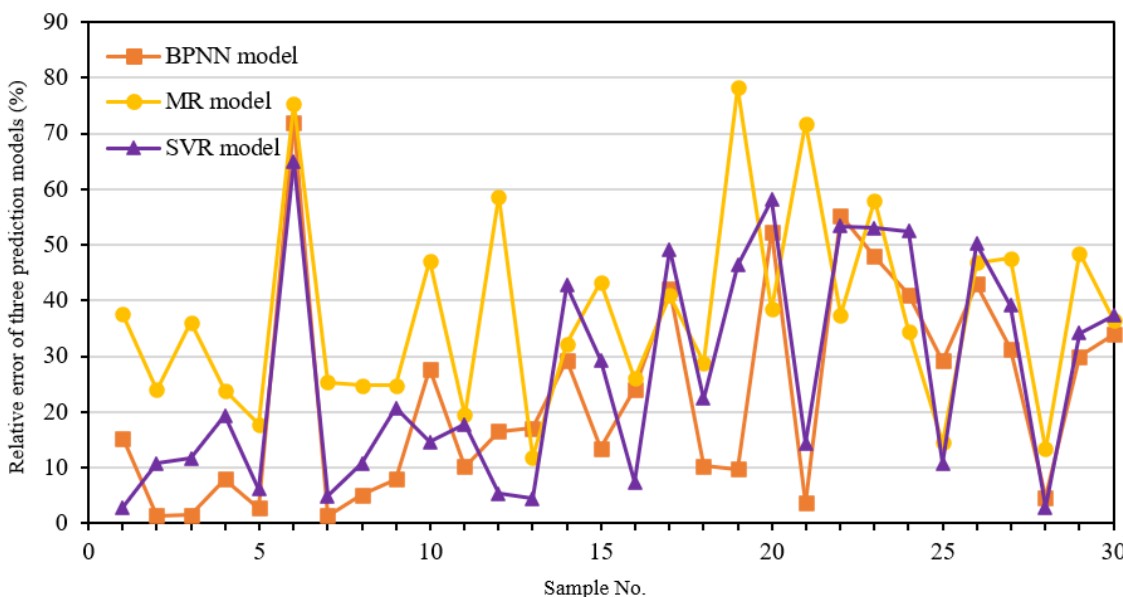

**Figure 16.** Contrast curve of relative error between predicted value and actual value of three prediction models. (The horizontal axis represents the number of thirty groups of samples in the test set).

It can be seen from Figure 15 that the absolute error of the prediction value of the multiple regression prediction model is the largest at 20 sample points, and the absolute error of 6 sample points is greater than 1 m/h, of which the absolute error of sample point 23 is the largest, reaching 1.45 m/h. Only 4 sample points have the smallest absolute error compared with the other two prediction models, of which the absolute error of sample point 13 is the smallest, reaching 0.15 m/h. There are only 13 points with an absolute error less than 0.5 m/h, and there are no sample points with an absolute error less than 0.1 m/h. The absolute errors of three sample points of the BPNN model are greater than 1 m/h, and the maximum absolute error of 1.27 m/h is obtained at sample point 22; compared with the other two prediction models, the absolute error of 18 sample points is the smallest. The absolute error of 5 sample points is less than 0.1 m/h, and the minimum absolute error of 0.02 m/h is obtained at sample point 3. The absolute error of 5 sample points of SVR model is greater than 1 m/h, and the maximum absolute error of 1.33 m/h is obtained at sample point 23. The absolute error of 8 sample points is the smallest compared with the other two prediction models. The absolute error of 5 sample points is less than 0.1 m/h. The minimum absolute error of 0.03 m/h is obtained in sample 12. Thus, compared with the multiple regression model and SVR model, the BPNN model has fewer sample points in the range of the absolute error greater than 1 m/h and more sample points in the range of the absolute error less than 0.1 m/h.

It can be seen from Figure 16 that among the predicted values of the multiple regression prediction model, the relative errors of 21 sample points are the largest, and the relative errors of 5 sample points are more than 50%. Among them, the relative error of sample point 19 is the largest, reaching 78.18%, and the relative errors of all sample points are more than 10%, and the minimum relative error of sample point 13 is 11.9%. The relative error of 5 sample points of the BPNN model is less than 5%, the minimum relative error is 1.42% at sample point 2 and the relative error of 3 sample points is greater than 50%. The relative error of 4 sample points of the SVR model is less than 5%, the minimum relative error is 2.82% at sample point 28, the relative error of 5 sample points is more than 50% and the maximum relative error is 58.15% at sample point 20. Thus, compared with the multiple regression model and SVR model, the BPNN model has fewer sample points in the range of a relative error greater than 50% and more sample points in the range of a relative error less than 5%. Therefore, compared with the multiple regression model and SVR model,

the BPNN prediction model has better prediction performance and generalization ability, especially in higher prediction accuracy and stability.

## 4. Conclusions

This paper takes machine learning as the main technical means, relying on the south of the Qinling tunnel of the Hanjiang-to-Weihe River Diversion Project to carry out the relevant research on TBM penetration analysis and prediction. Based on the field data, this paper analyzes and summarizes the influencing factors of the TBM performance of the deep-buried tunnel and uses the relevant methods of machine learning to establish the prediction model of the TBM penetration rate of the deep-buried tunnel. The main conclusions are as follows.

(1) Based on the establishment of the database of TBM performance in the south of the Qinling tunnel of the Hanjiang-to-Weihe River Diversion Project, the factors influencing TBM performance are quantitatively analyzed, including TBM mechanical parameters (cutterhead thrust (TF), cutterhead torque (T), cutterhead speed (RPM), etc.) and rock mass parameters (uniaxial compressive strength (UCS), the number of rock volume joints ($J_v$), angle between joint surface and tunnel axis ($\alpha$), etc.).

(2) Since there are many influencing factors on the TBM penetration rate, and the current situation is that the input parameters of the prediction model are determined artificially and the subjectivity is strong, the random forest (RF) algorithm is used to select the characteristics of the influencing factors on the TBM penetration rate, and the weight of the influencing factors is sorted to obtain the parameters of cutterhead thrust (TF), the number of rock volume joints ($J_v$), uniaxial compressive strength (UCS) and rotational speed (RPM). The total weight is 75.99%, which is comprised of the most important four parameters affecting the TBM penetration rate. The input parameters are determined for the TBM penetration rate prediction model based on machine learning.

(3) The prediction model of the TBM penetration rate based on multiple regression (MR) is established. The penetration index (FPI) is used as a bridge to connect the rock mass parameters and TBM mechanical parameters. The empirical formula of the TBM penetration rate prediction with independent variables of cutterhead thrust (TF), the number of rock volume joints ($J_v$), uniaxial compressive strength (UCS) and rotational speed (RPM) is obtained. The empirical formula can be applied to the preliminary prediction of the TBM penetration rate in the south of the Qinling tunnel of the Hanjiang-to-Weihe River Diversion Project.

(4) Based on the input parameters selected by the random forest algorithm, the prediction models of the TBM penetration rate are established based on the BP neural network (BPNN) and support vector regression (SVR). After training the training set and setting the parameters on the test set, the mean absolute percentage error (MAPE) and mean square error (MSE) of the BPNN prediction model are 22.95% and 0.3445, respectively. The mean absolute percentage error (MAPE) and mean square error (MSE) of the SVR model are 26.59% and 0.4085, respectively.

(5) Through the comparative analysis of the three prediction models, the BPNN prediction model shows better prediction performance and generalization ability than the multiple regression model and SVR model, which are embodied in higher prediction accuracy and stability.

**Author Contributions:** Writing—original draft, T.M., Y.J., Z.L. and Y.K.P. All authors have read and agreed to the published version of the manuscript.

**Funding:** This research was funded by the National Key Research Development Plan (No. 2018YFC1505301) and the Chinese National Natural Science Foundation (No. 41941018).

**Informed Consent Statement:** Informed consent was obtained from all subjects involved in the study.

**Conflicts of Interest:** The authors declare no conflict of interest.

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
