# Peer review of "Research on Prediction of TBM Performance of Deep-Buried Tunnel Based on Machine Learning"

_applsci, doi:10.3390/app12136599_

Round 1
Reviewer 1 Report
The abstract text is taken from the thesis and should be corrected.
The second line - this thesis ..
The vertical axis of Figure 2 is unclear.
The title of the horizontal and vertical axis of Figures is in Chinese and should be corrected.
This article is a copy of the text and needs to be completely rewritten.
Figures 13 and 14 have unclear content.
How to determine the fpi index should be described in Section 3.
Validation of research results is not clear.
What assumptions are considered in defining the bpnn model? These assumptions must be explained.
Specify the case study details in the research.
The number of pages of the article is large and the introduction and additional items should be removed and shortened.
Author Response
Point 1:The abstract text is taken from the thesis and should be corrected.The second line - this thesis ..
Response 1: Has been corrected.
Point 2:The vertical axis of Figure 2 is unclear.
Response 2: The vertical axis of Figure 2 has been modified again.
Point 3:The title of the horizontal and vertical axis of Figures is in Chinese and should be corrected.
Response 3:The title of the horizontal and vertical axis of Figures is be corrected.
Point 4:Figures 13 and 14 have unclear content.
Response 4: Figure 13 and 14 has been modified again.
Point 5:How to determine the fpi index should be described in Section 3.
Response 5: The uniaxial compressive strength UCS of rock mass and the number of rock volume joints Jv representing the integrity of rock mass will be used as independent variables in the multiple regression analysis of the penetration index FPI, and the empirical prediction formula based on the penetration index FPI will be established, namely formula (3.1).
Point 6:What assumptions are considered in defining the bpnn model? These assumptions must be explained.
Response 6: The error back propagation neural network continuously modifies the connection weights between each neuron through the training method of error back propagation. After each neuron connection calculation, the loss function on the output node can be represented by three loss functions: mean square error, mean absolute percentage error and mean absolute error.
Point 7:The number of pages of the article is large and the introduction and additional items should be removed and shortened.
Response 7: The introduction and additional items has be removed and shortened.

Reviewer 2 Report
In the manuscript the authors presented the influence of various factors on the performance of Tunned boring machine. Random forest algorithm and three prediction models were used: Multiple Regression model, Back Propagation Neutral Network Model and Support Vector Regression Model. The idea is interesting, especially it refers to the real construction process – Qinling tunnel of the Hanjiang-to-Weihe River Diversion Project.
However, in the reviewer’s opinion, the manuscript requires many changes and supplements before publication.
Main comments:
1) there is no explanation of what data is analyzed (there is no parameters values)
2) the models used are too briefly described and there is no reference to the literature
3) the analysis of the obtained results should be more detailed
Other comments:
1) English language needs to be improved. There are many mistakes and the terms used do not come from the scientific language
2) there are many editorial mistakes:
- no spacing
- no capital letters
- incorrect units, e.g. Mpa, kNM
- no explanation of symbols in the formulas (e.g. formula (3.2))
- explanations in Chinese on the charts
- no explanation of the vertical axes on the charts (Fig. 2)
- the horizontal axes are not described in detail (Figs. 4, 5, 6, 7,…)
- abbreviations are used without explanation (TBM – title, DTSS – line 65, CSM – line 47)
There are many editorial mistakes, and the reviewer recommends the authors to read the article carefully in this respect.
Finally, in the reviewer’s opinion, the article cannot be published in the Applied Sciences in this form, but after major revision it is worth considering the possibility of publication.
Author Response
Point 1: there is no explanation of what data is analyzed (there is no parameters values)
Response 1: Parameter value of data has been added in Table 2.
Point 2: the models used are too briefly described and there is no reference to the literature
Response 2: Model description and references have been added.
Point 3:the analysis of the obtained results should be more detailed
Response 3:
It can be seen from Figure 15 that the absolute error of the prediction value of the multiple regression prediction model is the largest at 20 sample points, and the absolute error of 6 sample points is greater than 1 m/h, of which the absolute error of sample point 23 is the largest, reaching 1.45m/h, and only 4 sample points have the smallest absolute error compared with the other two prediction models, of which the absolute error of sample point 13 is the smallest, reaching 0.15 m/h; There are only 13 points with absolute error less than 0.5m/h, and there are no sample points with absolute error less than 0.1 m/h. The absolute error of three sample points of BPNN model is greater than 1 m/h, and the maximum absolute error of 1.27 m/h is obtained at sample point 22; Compared with the other two prediction models, the absolute error of 18 sample points is the smallest. The absolute error of 5 sample points is less than 0.1m/h, and the minimum absolute error of 0.02 m/h is obtained at sample point 3. The absolute error of five sample points of SVR model is greater than 1 m/h, and the maximum absolute error of 1.33 m/h is obtained at sample point 23; The absolute error of 8 sample points is the smallest compared with the other two prediction models. The absolute error of 5 sample points is less than 0.1m/h. The minimum absolute error of 0.03 m/h is obtained in sample 12. Thus, compared with multiple regression model and SVR model, BPNN model has fewer sample points in the range of absolute error greater than 1 m/h and more sample points in the range of absolute error less than 0.1 m/h.
It can be seen from figure 16 that among the predicted values of the multiple regression prediction model, the relative errors of 21 sample points are the largest, and the relative errors of 5 sample points are more than 50%. Among them, the relative error of sample point 19 is the largest, reaching 78.18%, and the relative errors of all sample points are more than 10%, and the minimum relative error of sample point 13 is 11.9%. The relative error of 5 sample points of BPNN model is less than 5%, the minimum relative error is 1.42% at sample point 2, and the relative error of 3 sample points is greater than 50%. The relative error of four sample points of SVR model is less than 5%, the minimum relative error is 2.82% at sample point 28, the relative error of five sample points is more than 50%, and the maximum relative error is 58.15% at sample point 20. Thus, compared with multiple regression model and SVR model, BPNN model has fewer sample points in the range of relative error greater than 50% and more sample points in the range of relative error less than 5%.Therefore, compared with the multiple regression model and SVR model, the BPNN prediction model has better prediction performance and generalization ability, especially in higher prediction accuracy and stability.
Point 4: there are many editorial mistakes.
Response 4: Editorial mistakes have been modified.

Round 2
Author Response
N/A

Reviewer 2 Report
The authors revised and supplemented the manuscript in terms of content. Although the manuscript still contains editorial errors, e.g. unreadable, incorrect or incomplete axes in the Figures (1,2,6,8,9,11,12…). These errors make the article unprofessional and can be difficult to understand. The reviewer recommends reading the article carefully and correcting any mistakes.
Finally, in the reviewer’s opinion, the manuscript may be considered for publication in the Applied Sciences after revision made by authors.
Author Response
Point 1: The authors revised and supplemented the manuscript in terms of content. Although the manuscript still contains editorial errors, e.g. unreadable, incorrect or incomplete axes in the Figures (1,2,6,8,9,11,12…). These errors make the article unprofessional and can be difficult to understand. The reviewer recommends reading the article carefully and correcting any mistakes.
Response 1: The explanation of the horizontal axis is added in the title brackets in the Figures (1, 6, 7, 8, 9, 10, 11, 12, 13, 14, 15 and 16). Units(%) is added in the vertical axis.

This manuscript is a resubmission of an earlier submission. The following is a list of the peer review reports and author responses from that submission.